# A Decision Aid for Postpartum Adolescent Family Planning: A Quasi-Experimental Study in Tanzania

**DOI:** 10.3390/ijerph20064904

**Published:** 2023-03-10

**Authors:** Stella E. Mushy, Shigeko Horiuchi, Eri Shishido

**Affiliations:** 1Community Health Department, School of Nursing, Muhimbili University of Health and Allied Sciences, Dar es Salaam P.O. Box 65001, Tanzania; 2Women’s Health and Midwifery, School of Nursing Science, St. Luke’s International University, Tokyo 104-0044, Japan

**Keywords:** contraceptive uptake, decision aid, postpartum family planning, pregnancy, adolescents, Tanzania

## Abstract

Background: We evaluated the effects of our postpartum *Green Star* family planning decision aid on the decisional conflict, knowledge, satisfaction, and uptake of long-acting reversible contraception among pregnant adolescents in Tanzania. Methods: We used a facility-based pre–post quasi-experimental design. The intervention arm received routine family planning counseling and the decision aid. The control received only routine family planning counseling. The primary outcome was the change in decisional conflict measured using the validated decision conflict scale (DCS). The secondary outcomes were knowledge, satisfaction, and contraception uptake. Results: We recruited 66 pregnant adolescents, and 62 completed this study. The intervention group had a lower mean score difference in the DCS than in the control (intervention: −24.7 vs. control: −11.6, *p* < 0.001). The mean score difference in knowledge was significantly higher in the intervention than in the control (intervention: 4.53 vs. control: 2.0, *p* < 0.001). The mean score of satisfaction was significantly higher in the intervention than in the control (intervention: 100 vs. control: 55.8, *p* < 0.001). Contraceptive uptake was significantly higher in the intervention [29 (45.3%)] than in the control [13 (20.3%)] (*p* < 0.001). Conclusion: The decision aid demonstrated positive applicability and affordability for pregnant adolescents in Tanzania.

## 1. Introduction

Adolescent pregnancies remain a global public health concern with the highest rate occurring in developing countries. Most of these pregnancies are unintended [1,2], that is, unplanned or unwanted pregnancies [3]. The effective use of family planning is one of the most important and fundamental methods for preventing high-risk pregnancies, which typically happen too early and frequently [4]. A high-impact intervention that lowers the risks of adolescent pregnancy is the use of long-acting reversible contraception (i.e., intrauterine copper devices and implants) right away following childbirth [5]. Studies reported that adolescents who started long-acting reversible contraception following childbirth had a lower chance of recurring unwanted pregnancies than adolescents who started short-acting reversible contraception or who did not use any family planning techniques [6].

Despite substantial improvements in the education of healthcare professionals, the provision of quality care, and the distribution of family planning supplies, postpartum contraceptive usage among adolescents in Tanzania continues to be woefully underused. Only 12.2% of adolescent mothers use postpartum contraception within three months of childbirth, with injectables being the most popular method followed by pills [7]. Pregnant adolescents require thorough information on all the contraceptive methods that are available for use immediately following childbirth in their country in order to make an informed choice about which method suits them. Intrauterine copper devices and implants are the only long-acting reversible contraceptive options available immediately after childbirth in Tanzania. Both intrauterine copper devices and implants are quite effective at preventing pregnancy, and they all have a long lifespan and are simple to use [5,8]. So, clarifying values, beliefs, and priorities is necessary in the case of pregnant adolescents to help them make the decision to use the method that suits them.

Decision aids have been beneficial, as they inform and educate patients about the available treatment options, which helps reduce decisional conflicts [9]. Patient decision aids are the resources that make it easier for patients to participate in decision-making by outlining the choice that must be made, outlining the possibilities and potential results, and defining personal values [10]. Recently, decision aids are utilized to inform patients and the general public about health issues [11]. Research has demonstrated the effectiveness of decision aids in family planning counseling [12,13]. To our knowledge, however, there has been limited research on decision-support tools that emphasize long-acting reversible contraceptive techniques to increase family planning uptake by pregnant adolescents right after childbirth. This study designed [14] and tested the efficacy of the postpartum “*Green Star*” family planning decision aid on the decisional conflict, knowledge, satisfaction, and uptake of contraception among pregnant adolescents in Tanzania. We hypothesized that pregnant adolescents using the postpartum “*Green Star*” family planning decision aid will have a lower decision conflict scale (DCS) score than pregnant adolescents who are not using the decision aid (control group) and that knowledge and satisfaction scores will be higher in pregnant adolescents using the decision aid than in pregnant adolescents who are not using the decision aid (control group).

## 2. Materials and Methods

### 2.1. Study Design, Setting, and Participants

This study used a pre–post quasi-experimental design with concurrent control. The study area was Pwani region, which is among the regions in Tanzania with the highest childbearing rates. Thirty percent (30%) of pregnant women in the Pwani region are women aged 15–19 years [7]. Mkuranga and Kisarawe are among the 6 districts of the Pwani region in Tanzania. The selected district hospitals are government-run public health facilities that also serve low-income populations.

The inclusion criteria were pregnant adolescents who were between 15–19 years (considered adults in Tanzania) in their 28 gestation weeks planning to deliver at the hospital where they are attending antenatal care services and who were willing and provided consent to participate (Appendix A). The exclusion criteria were pregnant adolescents who were receiving family planning education from other programs. If the participants were minors (10–14 years), informed consent that included the signature of their parents/guardian was required (Appendix A). Participants could also withdraw from this study at any time (Appendix A). The legal age at which a young person in Tanzania may obtain contraceptive services is undefined, and parental consent is not required (Appendix A [page 5]). Thus, there is no clear law that prevents the use of contraceptives at any age. In addition, marriage law in Tanzania allows a girl who has attained an apparent age of 15 years to get married based on the Law of Marriage Act in Tanzania.

Adolescents were selected as the study participants because they have a high risk of rapid repeat pregnancy (RRP), and they critically underutilize different types of modern contraception compared with women above 20 years [15].

### 2.2. Sampling, Training, and Sample Size

Consecutive sampling was used to select the study participants. Enrollment of this study’s participants in both groups ran concurrently, and the study sites were in different districts, which reduced data contamination risk. Two RAs (AM and CM) who were certified midwives recruited the participants and conducted this study in close coordination with the lead researcher (SM). The criteria for selecting the RAs were that they were experienced in family planning counseling, able to conduct intervention studies, and working in an antenatal clinic. Each RA received two days of training separately to avoid data contamination.

The sample size was determined based on a systematic review by Stacey et al. [16] The data included all published studies that used a randomized controlled trial design evaluating patient decision aids. The data showed a mean difference in the decision conflict scores of −7.22 [−9.12, −5.31]; the estimated sample size needed in each group was 29 participants. In the present study, we also considered a dropout rate of about 10% and thus set the number of participants to 32 in each group for a total of 64 participants recruited (effect size 0.5; power 0.8; significance level 5%).

### 2.3. Ethical Considerations

The Research Ethics Boards of St. Luke’s International University (Appendix A; Approval number: 20-A91; Approval date: 19 March 2021), Muhimbili University of Health and Allied Sciences (Appendix A; Approval number: MUHAS-REC-1-2020-076; Approval date: 4 March 2020) and National Institute of Medical Research approved this study. This study was registered in the Clinical Trials Registry of University Hospital Information Network in Japan (UMIN000028471).

### 2.4. Intervention

#### 2.4.1. Development of the Postpartum “*Green Star*” Family Planning Decision Aid

We developed the postpartum “*Green Star*” family planning decision aid by initially identifying the research gap, target population, and study objectives to address the research problem on adolescent pregnancy. We carefully focused and examined previously published studies when determining the study objectives [11,13,17,18].

Thereafter, we identified the individual needs of the participating pregnant adolescents by reviewing a previous study that looked into barriers to the utilization of family planning among female youths in Dar es Salaam, Tanzania [19].

The content, design, and arrangement of the developed prototype decision aid were based on the Ottawa Patient Decision Aid Development eTraining [10], International Patient Decision Aid Standards Collaboration Checklist [20], theory of planned behavior [21], health belief model [22], social cognitive theory [23], current clinical guides for family planning counseling for providers [1,8], and findings from previous studies on the benefits and side effects of the different options, satisfaction and continuation rates, and fertility return [24,25,26,27,28,29,30]. The prototype decision aid has four components based on the Ottawa Patient Decision Aid Development eTraining guide: (i) know how to make a decision with conviction; (ii) understand the characteristics of the decision; (iii) clarify what is important to you; and (iv) make the decision. We shared the prototype decision aid with three experts, namely, our research supervisor and two midwives, all with extensive years of experience in maternal and child health and in developing decision aids. The aim of sharing the prototype decision aid was to receive comments on the comprehensibility and usefulness of the prototype decision aid, which we incorporated in the modified and improved postpartum “*Green Star*” family planning decision aid.

We then carried out a feasibility study of the prototype decision aid to assessing its practicality, usefulness, and acceptability as perceived by pregnant adolescents and healthcare providers [14]. Finally, we developed the third and final version of the decision aid and named it postpartum “*Green Star*” family planning decision aid. This decision aid was then assessed for its effects on decisional conflict, knowledge, satisfaction, and uptake of long-acting reversible contraception (Appendix A).

#### 2.4.2. Intervention Group

Each participant in the intervention group first received the routine family planning counseling offered by a healthcare provider on duty that day. Then, this was followed by individual face-to-face family planning counseling education using the contents of the postpartum “*Green Star*” family planning decision aid. It took about 30–40 min for the research assistant (RA) to present all the contents to each participant. Every participant received the 10-page postpartum “*Green Star*” family planning decision aid and brought it home for further reading and reference when needed. The participants received three education sessions at different times before giving birth (Figure 1). Three education sessions were considered sufficient for the participants to understand the methods and address all the barriers preventing them from using family planning.

#### 2.4.3. Control Group

The participants in the control group only received routine family planning counseling offered at each antenatal clinic visit. Each participant received three education sessions as in the intervention group (Figure 1).

### 2.5. Data Collection Methods

#### 2.5.1. Primary Outcome

Decisional conflict was assessed at Time 1 (28 gestational weeks) and at Time 4 (within 2 days after childbirth) using the DCS. The DCS is a 10-item self-report questionnaire that measures a patient’s uncertainty about what treatment option to choose and the factors associated with the uncertainty (e.g., lack of information, myths and misconception, and lack of support) [31]. The DCS has 4 subscales: informed, clarity, support, and uncertainty). Informed was conceptualized as having clear information about the available long-active reversible contraception, including how they work to prevent pregnancy, their benefits, and side effects. Clarity was defined as the quality of someone being clear and easy to understand about personal values for benefits and risks/side effects of long-acting reversible contraception. Support was hypothesized as the feeling a participant had regarding the assistance she receives from healthcare providers and her significant others in deciding to use long-acting reversible contraception. Uncertainty was defined as the feeling of not being completely confident or sure of the best choice of long-acting reversible contraception. Items were answered using a 3-point Likert scale (0 = “Yes”; 2 = “unsure”; 4 = “no”) with scores ranging from 0 to 100. A higher score meant a higher decisional conflict and uncertainty and vice versa [31]. As we adapted items in the DCS, we conducted Cronbach’s Alpha test to check for internal consistency of the DCS items, and we obtained 0.848, which was higher than the commonly recommended value of 0.6. The results indicated that a set of items in the DCS were reliable (i.e., closely related as a group).

#### 2.5.2. Secondary Outcomes

The secondary outcomes were knowledge, satisfaction with decision-making, and uptake of long-acting reversible contraception. Knowledge was assessed at Time 1 and Time 3 (36–38 gestational weeks). We created a knowledge questionnaire to test the participants’ knowledge of long-acting reversible contraception. We adapted questions from different reports [8,19,24,25,28,29,32], which included advantages, disadvantages, side effects, myths, and misconceptions commonly existing in the community. The knowledge questionnaire had 7 questions, each with a score of 1 for a total of 7 scores. The questions required a “yes” or “no” response. A score of 7 indicated that a participant is knowledgeable about long-acting reversible contraception and vice versa. Satisfaction was only assessed at Time 4. We assessed satisfaction using an effective decision-making subscale of the DCS that had 4 items with 3 responses (4 = “yes”, 2 = “unsure”, 0 = “no”) with scores ranging from 0 to 100. A higher score indicated a higher satisfaction with decision-making and vice versa. The decision on which option to use following childbirth was only assessed at Time 4 using 1 question that asked, “Which option do you prefer?”.

#### 2.5.3. Demographic Data

The information collected as part of the questionnaire included age, parity, highest education level, occupation, and marital status.

### 2.6. Data Analysis

Data were descriptively analyzed using IBM SPSS Statistics version 24.0. Descriptive analysis was used to analyze the demographic information of the participants to determine frequencies and percentages of their distribution within groups. The chi-square test was conducted to analyze data and to observe the distribution within the group for the ordinal and between groups for categorical data. The analysis was based on calculating the mean score differences of the selected variables at Time 1, Time 2, Time 3, and Time 4 between groups. The independent sample t-test was performed to compare the mean scores of DCS, knowledge, and satisfaction between groups. Multiple linear regression to predict the DCS score and logistic regression analysis for long-acting reversible contraception uptake were performed. The statistical tests were performed with a two-sided 5% level of significance.

## 3. Results

### 3.1. Flow of This Study

Data were collected for 7 months from early March to the end of September 2021. The participant flow diagram is shown in Figure 1. A total of 80 pregnant adolescents were eligible, but only 66 gave their written informed consent and were included in this study with 33 participants in each group. Two participants, one from the intervention and one from the control group were lost to follow-up and were excluded from the analysis. Therefore, a total of 64 participants were included for analysis, each group with a total of 32 participants.

### 3.2. Sociodemographic Characteristics of the Study Participants

Details on the sociodemographic information of the study participants are shown in Table 1. The mean age of the study participants in the intervention group was 17.5 (1.29) years, whereas that in the control group was 18.0 years (SD 0.71) [age range, 15–19 years]. Marital status and occupation showed a significant difference between the two groups (*p* < 0.001). The ratio of single pregnant adolescents was higher in the intervention group [23 (71.9%)] than in the control group [7 (21.9%)]. Similarly, the ratio of employed pregnant adolescents was higher in the intervention group [27 (84.4%)] than in the control group [8 (25%)].

### 3.3. Decisional Conflict

The total DCS mean score at Time 1 was significantly lower in the intervention group than in the control group (intervention group: 65.00 [SD 19.1] vs. control group: 77.80 [SD 18.4], t = −2.72, *p* = 0.008) (Table 2). The observed mean scores in the DCS subscales “Support” (*p* = 0.723) and “Uncertainty” (*p* = 0.548) were not significantly different between the two groups at Time 1. However, at Time 4, there was a significant difference in the total DCS mean score between the intervention group and the control group (intervention group: 3.13 [SD 4.7] vs. control group: 48.5 [SD 29.6], t = −8.55, *p* < 0.001). Additionally, the mean scores of all four subscales (i.e., informed, clarity, support, and uncertainty) in the DCS between the two groups at Time 4 showed a significant difference (*p* < 0.001). The mean score difference in the DCS (Time 4 minus Time 1) was significantly lower in the intervention group than in the control group (intervention group: −24.7 [SD 7.99] vs. control group: −11.6 [SD 10.9], t = −5.53, *p* < 0.001). The mean score difference of all four subscales in the DCS was significantly lower in the intervention group than in the control group. These results support the hypothesis that the decision-making conflict score will be lower in the intervention group than in the control group.

### 3.4. Multiple Linear Regression for Predicting DCS Score

Multiple linear regression was performed to predict the DCS score based on age, occupation, and marital status at Time 1, Time 4, and at the time differences (Time 4 minus Time 1). Age was the only variable that showed a significant relationship in the control group both at Time 1 (*β* = 0.455, *p* = 0.015) and at Time 4 (*β* = 0.506, *p* = 0.006). However, age did not show a significant relationship at the time differences (Time 4 minus Time 1). As for the intervention group, none of the variables showed a significant relationship with the DCS scores (Table 3).

### 3.5. Contraceptive Knowledge, Satisfaction, and Uptake

At Time 1, the results showed no significant difference in the knowledge mean score between the two groups (intervention group: 1.84 [SD 1.98] vs. control group: 2.34 [SD 1.61], t = −1.1, *p* = 0.274). At Time 3, the knowledge mean score was significantly higher in the intervention group than in the control group (intervention group: 6.38 [1.60] vs. control group: 4.34 [1.82], t = 4.733, *p* < 0.001). The mean score difference in knowledge (Time 3 minus Time 1) in the intervention group was larger than that in the control group (intervention group: 4.53 [2.54] vs. control group: 2.00 [1.45], t = 4.88, *p* < 0.001). These findings support the hypothesis that the level of knowledge of long-acting reversible contraception will be higher in the intervention group than in the control group.

The mean score of satisfaction was significantly higher in the intervention group than in the control group (intervention group: 100.00 [SD 0.0] vs. control group: 55.8 [SD 30.7], t = 8.112, *p* < 0.001). The proportion of “yes” responses to each item about satisfaction was greater in the intervention group (100%) than in the control group (< 40%). The chi-square test result of each item between the two groups was found to be significant in all items (*p* < 0.001). These results support the hypothesis that the state levels of satisfaction with decision-making will be higher in the intervention group than in the control group.

The proportion of participants who decided to use long-acting reversible contraception showed significant differences between the two groups. The proportions of participants who “did not decide to use any option” were 3 (4.7%) in the intervention group and 19 (29.7) in the control group. However, the proportions of participants who “decided to use implant” were 29 (45.3%) in the intervention group and 13 (20.3%) in the control group (x^2^ = 17.73, *p* < 0.001). These findings support the hypothesis that the proportion of long-acting reversible contraception uptake will be higher in the intervention group than in the control group.

### 3.6. Logistic Regression Analysis for Long-Acting Reversible Contraception Uptake

Age, marital status, and occupation showed no significant relationship with long-acting reversible contraception uptake in the intervention group. However, age alone showed a significant relationship with long-acting reversible contraception uptake in the control group with a negative *β* value (Table 4).

## 4. Discussion

### 4.1. Decisional Conflict for Pregnant Adolescents

This study assessed the effect of the developed postpartum family planning decision aid on the decisional conflict, knowledge, satisfaction, and uptake of long-acting reversible contraception among pregnant adolescents. The study results showed that participants in the intervention group had a lower decisional conflict, higher knowledge, higher satisfaction, and higher contraceptive uptake prevalence than participants in the control group. The present study is the first study in Tanzania using decision aids related to adolescent use of long-acting reversible contraception. Although decision aids have been used in a range of other study areas, some findings concurred with the present study’s findings, whereas others did not.

The present results concur with the results of a systematic review that involved 105 studies on decision aids from different areas, namely, cancer screenings, prenatal complication diagnosis, immunizations, and diabetic treatments [9]. These studies found that the use of decision aids more markedly reduced the mean difference in the DCS score in the intervention group (with decision aids) than in the control group (without decision aids) (MD −9.28, 95% CI: −12.2 to −6.36). The effect of decision aids on medication choice for diabetes mellitus was also assessed in three randomized clinical trials [33,34]. The findings of these trials showed a significantly lower mean difference in the DCS score in the intervention group than in the control group. Two clinical trials involving pregnant women have also been conducted [17,35]. The first clinical trial assessed the effect of decision aids on the choice of pregnant women whether to have epidural anesthesia or not during labor [17]. The second randomized clinical trial evaluated the effect of decision aids on women with breech presentation at term [35]. The findings of the two previous clinical trials were similar to the findings of the present study in that the mean difference in the DCS score was significantly lower in the intervention group which received a decision aid than in the control group which received only standardized routine care.

On the other hand, the present findings are inconsistent with the findings from a previous study that evaluated the effect of a decision aid on decision-making for the treatment of pelvic organ prolapse [36]. The previous study found that the DCS score of patients who received a decision aid and standard counseling was not significantly lower than the DCS score of patients who received only standard counseling (*p* = 0.566). The probable reasons include the already available pelvic organ prolapse decision aid in the setting and the regular review of information by the patients together with the clinician at the initial encounter.

In the present study, a significant mean difference was observed in the intervention group because there are no family planning decision aids in antenatal clinics to help patients decide on the family planning option that they would take. A study on the experiences of diabetic patients and healthcare providers on shared decision-making conducted in Tanzania found that neither the patients nor the healthcare providers had been using decision-making aids; the patients reported that only health education tools are being used for educating them [37].

The multiple regression analysis of the DCS score based on age, marital status, and occupation showed age as the only variable that had a significant relationship with the DCS score in the control group at Time 1 and Time 4 but not in the intervention group. The findings further show that as age increases, the decision conflict score also increases and vice versa. These findings indicate that if younger adolescents receive the correct information about long-acting reversible contraception at the right time, this will improve their chances of utilizing family planning methods. Regarding the lower DCS score in the intervention group, the present findings suggest that the decision aid played an important role in imparting knowledge and correcting the held myths and misconceptions of the younger adolescents.

### 4.2. Knowledge, Satisfaction, and Uptake of Long-Acting Reversible Contraception for Pregnant Adolescents

The mean score difference of knowledge, satisfaction, and uptake of long-acting reversible contraception was significantly higher in the intervention group than in the control group. In a systematic review [9] of 52 studies, 4 randomized control trials [33,34,35,38] and 1 survey [39] found a significant increase in the scores of knowledge, satisfaction with decision making, and choice for the treatment options in the intervention group which used a decision aid compared with the control group which used standardized routine care.

Although the present study showed an increase in contraception uptake, none of the long-acting reversible contraception users chose an intrauterine copper device, as all the participants chose only implants. The main reasons for not using an intrauterine copper device might be related to individual perception factors, such as myths, misconceptions, and discomfort from postpartum pain [19]. This nonusage was related to the fear of expulsion and the risk of infection. If an intrauterine copper device is inserted immediately after childbirth and infection prevention control measures are adhered to, then the risk of expulsion and infection would be minimal. However, the postpartum “*Green Star*” family planning decision aid did not have any information regarding the timely insertion of an intrauterine copper device. In future studies, this information will be included in the decision aid to improve intrauterine copper device uptake.

Findings from the logistic regression analysis showed that the marital status and occupation variables did not have a significant relationship with knowledge and satisfaction in both groups. Only the age variable showed a significant relationship with the uptake of long-acting reversible contraception in the control group. The study participants in the control group were 21% (odds = 0.21) less likely to use long-acting reversible contraception than the study participants in the intervention group. Younger adolescents in the control group were more likely to utilize LARC than older adolescents. These findings inform that if family planning programs put great efforts into pregnant adolescents by ensuring that they get the right information at the right time about postpartum family planning, uptake will be improved following childbirth. In addition, we will plan to conduct a qualitative study using interviews focusing on the social–cultural context of adolescent pregnant mothers related to the decision aid of long-acting reversal contraception.

### 4.3. Strengths and Limitations

To our knowledge, this is the first study in Tanzania that used a postpartum family planning decision aid to assist in the decision-making of pregnant adolescents regarding which long-acting reversible contraception they should use following childbirth. This means the open practicality of using a decision aid for healthcare even though the subjects were adolescents. To avoid data contamination between groups, the hospital in the intervention group was located in a different district from the hospital in the control group, that is, 2–3 h of driving using private transport. On the other hand, the present findings cannot be generalized unless a randomized controlled study is conducted.

## 5. Conclusions

To our knowledge, this is the first quasi-experimental study with a control that evaluated the effects of our recently developed postpartum “*Green Star*” family planning decision aid on pregnant adolescents’ choice of using long-acting reversible contraception. The decision aid significantly lowered decision-making conflict, improved knowledge and satisfaction with decision-making, and enhanced the uptake of available long-acting reversible contraception. The overall findings indicate the usefulness of the postpartum “*Green Star*” family planning decision aid, as it supplemented and supported patient–provider communications during family planning counseling in antenatal clinics.

## Figures and Tables

**Figure 1 ijerph-20-04904-f001:**
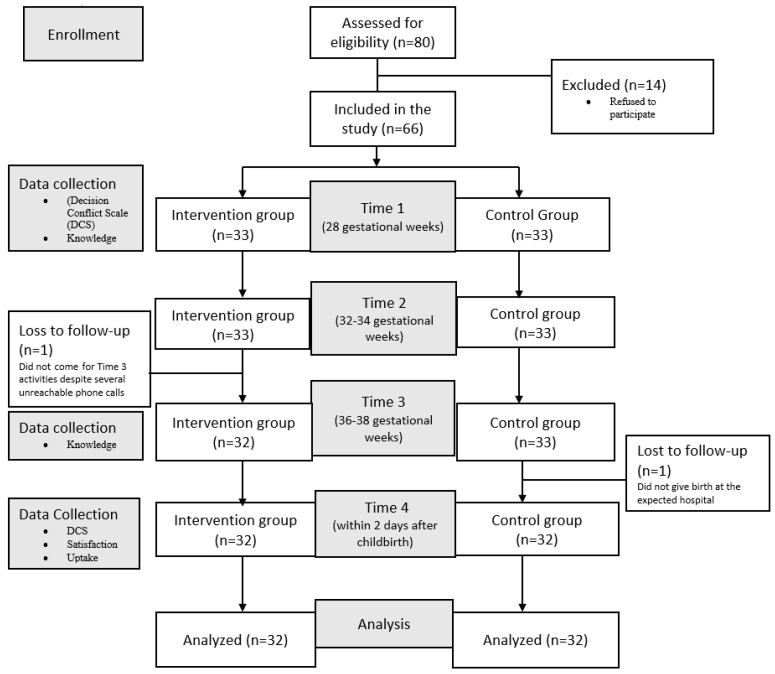
Flow diagram of study participants.

**Table 1 ijerph-20-04904-t001:** Sociodemographic characteristics of the study participants.

	Intervention Group (n = 32)	Control Group (n = 32)	t	x^2^	*p*-Value
Age (years)	17.5 (SD 1.29)	18.0 (SD 0.71)	−2.15		0.03
mean (SD)					
Marital status					
Single	23 (71.9%)	7 (21.9%)		19	<0.001
Married	9 (28.1%)	17 (53.1%)			
Cohabiting	0 (0.0%)	8 (25.0%)			
Gravidity					
1	31 (96.9%)	29 (90.6%)		1.07	0.302
2	1 (3.1%)	3 (9.4%)			
Highest education level					
Primary	27 (84.4%)	27 (84.4%)		0.48	0.788
Secondary	3 (9.4%)	4 (12.5%)			
None	2 (6.2%)	1 (3.1%)			
Occupation					
Employee	27 (84.4%)	9 (28.1%)	45.87		<0.001
Housewife	1 (3.1%)	20 (62.5%)			
Do not work	4 (12.5%)	3 (9.4%)			

**Table 2 ijerph-20-04904-t002:** Mean Scores of the Decisional Conflict Scale.

			Intervention Group (n = 32)	Control Group (n = 32)	t	*p*-Value
			Mean (SD)	Mean (SD)		
Time 1	Total	DCS (0–100)	65.00 (SD 19.1)	77.80 (SD 18.4)	−2.72	0.008
	Subscale	Informed	65.6 (SD 23.9)	86.9 (SD 21.0)	−3.79	< 0.001
		Clarity	72.6 (SD 21.4)	91.4 (SD 21.6)	−3.48	0.001
		Support	54.6 (SD 23.2)	16.7 (SD 23.5)	−0.36	0.723
		Uncertainty	85.9 (SD 15.4)	88.2 (SD 15.5)	−0.60	0.548
Time 4	Total	DCS (0–100)	3.13 (SD 4.70)	48.5 (SD 29.6)	−8.55	<0.001
		Informed	4.17 (SD 8.46)	49.4 (SD 31.5)	−7.85	< 0.001
	Subscale	Clarity	8.59 (SD 13.6)	53.1 (SD 39.5)	−6.02	< 0.001
		Support	0.52 (SD 2.9)	47.9(SD 27.6)	−9.63	< 0.001
		Uncertainty	0.00(SD 0.00)	43.7(SD 37.0)	−6.68	< 0.001
Differences Time 4−Time 1	Total	Differences	−24.7 (SD 7.99)	−11.6 (SD 10.9)	−5.53	<0.001
	Subscale	Informed	−7.00 (SD 3.07)	−4.00 (SD 4.03)	−3.21	0.002
		Clarity	−6.00 (SD 2.09)	−4.00 (SD 3.21)	−3.04	0.004
		Support	−7.00 (SD2.78)	0.00 (SD 2.83)	−7.76	< 0.001
		Uncertainty	−6.00 (SD2.20)	−2.00 (SD 2.87)	−5.71	< 0.001

**Table 3 ijerph-20-04904-t003:** Multiple linear regression for predicting DCS score.

	Study Arms	Items	Beta	*p*-Value
Time 1	Intervention group	Age	0.292	0.12
		Occupation (dummy)	−0.115	0.532
	Control group	Age	0.455	0.015
		Occupation (dummy)	−0.308	0.09
Time 4	Intervention group	Age	−0.227	0.217
		Occupation (dummy)	0.305	0.099
	Control group	Age	0.506	0.006
		Occupation (dummy)	−0.092	0.597
Time 4-Time 1	Intervention group	Age	−0.333	0.073
		Occupation (dummy)	0.182	0.317
	Control group	Age	0.240	0.211
		Occupation (dummy)	0.108	0.572

**Table 4 ijerph-20-04904-t004:** Logistic regression analysis for long-acting reversible contraception uptake.

Study Arms	Item	Beta	*p*-Value	Odds	95% CI	
					Lower	Upper
Intervention group	Age	−0.56	0.45	0.57	0.13	2.5
	Marital status	−19.52	1.00	0.00	0.00	
	Occupation	−20.22	1.00	0.00	0.00	
Control group	Age	−1.55	0.03	0.21	0.05	0.84
	Marital status	−0.62	0.32	0.53	0.16	1.82
	Occupation	0.08	0.87	1.09	0.38	3.17

## Data Availability

These data cannot be made publicly available since they contain sensitive information about the participants, and also the participants did not provide their approval for the sharing of their interviews. For researchers who meet the requirements for access to confidential data, data are available from the Directorate of Research and Publications at the Muhimbili University of Health and Allied Sciences (contact via drp@muhas.ac.tz, Tel.: +2552150302-6).

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
