# Peer review of "A Decision Aid for Postpartum Adolescent Family Planning: A Quasi-Experimental Study in Tanzania"

_ijerph, 2023, doi:10.3390/ijerph20064904_

Round 1

Reviewer 1 Report

The authors present an interesting attempt to improve counselling of adolescent mothers about contraception post-partum.  Using a structured family planning decision aid, they analyzed the effects of the women’s knowledge and satisfaction as well as their decisional conflicts. This seems to be the first study on this field in Tanzania where are up to 30% rate of adolescent pregnancies. Therefore, it meet a major important health issue not only in that region.

The study design is well planned and it could be realized per protocol.

The statistical analysis is appropriate to adjust the imbalances caused by the non-randomized trial design.

The reported results demonstrate clearly the superiority of the family planning decision aid not only in regard of the primary and secondary outcomes but also in the “hard fact” of more than twofold increase of long-acting contraceptives in the better counselled women.

So the reviewer support to publish that manuscript.

Author Response

Thank you for your valuable comments.

Reviewer 2 Report

Thank you for submitting your work as a potential publication in the International Journal of Environmental Research and Public Health. Overall, the manuscript is sound, easy to read, and it focuses on a topic of paramount importance. My congratulations to the authors for the work developed. There are some scattered aspects that could be improved but, overall, the manuscript is well written and adequately addressed. The only important drawback that I perceive after a careful review is the sample size. In fact, the Central Limit Theorem does not state itself a minimum of 30 participants in each group: this information is somehow skewed, since it has been taken from a webpage that, in turn, stems from a paper by Chang, H. J., K. Huang, and C. Wu. ("Determination of sample size in using central limit theorem for weibull distribution." International Journal of Information and Management Sciences, Vol. 17, No. 3. 2006, pp. 153-174) that does not support or endorse the n=30 as a general aplicable rule, but quite the contrary, based on the fact that this generalized criterion may not be suitable for all kind of distributions. This fact, therefore, should be corrected, and the justification of the sample size should be differently focused, calculated and justified. Thank you.

Author Response

Point-by-Point Point Responses

  1. Comments

The only important drawback that I perceive after a careful review is the sample size. In fact, the Central Limit Theorem does not state itself a minimum of 30 participants in each group: this information is somehow skewed, since it has been taken from a webpage that, in turn, stems from a paper by Chang, H. J., K. Huang, and C. Wu. ("Determination of sample size in using central limit theorem for weibull distribution." International Journal of Information and Management Sciences, Vol. 17, No. 3. 2006, pp. 153-174) that does not support or endorse the n=30 as a general applicable rule, but quite the contrary, based on the fact that this generalized criterion may not be suitable for all kind of distributions. This fact, therefore, should be corrected, and the justification of the sample size should be differently focused, calculated and justified. Thank you.

Response:

Thank you for your valuable comments. We revised the explanation based on the other sample size calculation. Please see Lines 101-106, Page 3 in the attached file.

“The sample size was determined based on a systematic review by Stacey et al. [16] The data included all published studies that used a randomized controlled trial design evaluating patient decision aids. The data showed a mean difference in the decision conflict scores of -7.22 [-9.12, -5.31]; the estimated sample size needed in each group was 29 participants. In the present study, we also considered a dropout rate of about 10% and thus set the number of participants was set to 32 in each group for a total of 64 participants recruited (effect size 0.5; power 0.8; significance level 5%).”

Reference

“16. Stacey D, Légaré F, Lewis K, Barry MJ, Bennett CL, Eden KB, Holmes-Rovner M, Llewellyn-Thomas H, Lyddiatt A, Thomson R, Trevena L. Decision aids for people facing health treatment or screening decisions. Cochrane Database Syst Rev. 2017 Apr 12;4(4):CD001431. doi: 10.1002/14651858.”

Reviewer 3 Report

Thank you for the opportunity to review this paper, titled “A decision aid for postpartum adolescent family planning: A quasi-experimental study in Tanzania”. This is a well written paper and provides a a clear overview of the results of your quasi-experimental study that evaluated the effect of the postpartum Green Star family planning decision aid on decisional conflict, knowledge, satisfaction, and contraception uptake. Overall, I found the paper well structured with a clear abstract, well written Background and clearly described Methodology and Results. There are just a few areas where some further information or content would be useful.

Under Materials and Methods, on page 5, lines 166-167 – I can see that a reference is provided for the 4 subscales but a little more detail on what “Informed, Clarity, Support and Uncertainty” actually mean would be useful. This would then assist in understanding the results presented in Table 2.

 Discussion – it would be useful to start the discussion by restating the study aim and briefly highlighting the main results. It would then be worth stating that while other studies have not been conducted using decision aids related to adolescent use of long-acting contraceptive techniques, they have been used in a range of other areas, some of which concur with the findings of the present study and some which do not.

There are some results which are worthy of further discussion. In Table 1, differences between the Intervention and Control Group are shown in regard to the following areas: there are more single women in the Intervention Group compared to the Control Group (71.9% vs 21.9%); less women in the Intervention Group are married (28.1% vs53.1%); more women in the Intervention Group are working (84.4% vs 28.1%); and less women in the Intervention Group were housewives (3.1% vs 62.5%). These results are interesting, and I wonder if you may be able to discuss these in relation to the cultural context of these differences.

Author Response

Point-by-Point Point Responses

1.Comments

Under Materials and Methods, on page 5, lines 166-167 – I can see that a reference is provided for the 4 subscales but a little more detail on what “Informed, Clarity, Support and Uncertainty” actually mean would be useful. This would then assist in understanding the results presented in Table 2.

Response:

Thank you for your suggestion. We added the definition of each subscale as proposed. Please refer to Lines 169-177, Page 5.

“Informed was conceptualized as having clear information about the available long-active reversible contraception, including how they work to prevent pregnancy, their benefits, and side effects. Clarity was defined as the quality of someone being clear and easy to understand about personal values for benefits and risks/side effects of long-acting reversible contraception. Support was hypothesized as the feeling a participant had regarding the assistance she receives from healthcare providers and her significant others in deciding to use long-acting reversible contraception. Uncertainty was defined as the feeling of not being completely confident or sure of the best choice of long-acting reversible contraception.”

 2.Comments

Discussion – it would be useful to start the discussion by restating the study aim and briefly highlighting the main results. It would then be worth stating that while other studies have not been conducted using decision aids related to adolescent use of long-acting contraceptive techniques, they have been used in a range of other areas, some of which concur with the findings of the present study and some which do not.

Response:

We acknowledge your valuable suggestion. We added a paragraph describing the study aims and a summary of the study findings at the start of the Discussion as suggested. Please refer to Lines 292-300, Pages 8-9.

“The study assessed the effect of the developed postpartum family planning decision aid on decisional conflict, knowledge, satisfaction, and uptake of long-acting reversible contraception among pregnant adolescents. The study results showed that participants in the intervention group had a lower decisional conflict, higher knowledge, higher satisfaction, and higher contraceptive uptake prevalence than participants in the control group. The present study is the first study in Tanzania using decision aids related to adolescent use of long-acting reversible contraception. Although decision aids have been used in a range of other study areas, some findings concurred with the present study's findings, whereas others did not.”

3.Comments

There are some results which are worthy of further discussion. In Table 1, differences between the Intervention and Control Group are shown in regard to the following areas: there are more single women in the Intervention Group compared to the Control Group (71.9% vs 21.9%); less women in the Intervention Group are married (28.1% vs53.1%); more women in the Intervention Group are working (84.4% vs 28.1%); and less women in the Intervention Group were housewives (3.1% vs 62.5%). These results are interesting, and I wonder if you may be able to discuss these in relation to the cultural context of these differences.

Response:

Thank you for your important advice. We added explanations about results from the marital status and occupation variables using the logistic analysis in the Discussion section. In addition, we explained that we will have to grasp the cultural context using qualitative studies by interviewing adolescents in future research. Please see Lines 362-373, Page 10 in the attached file.

“Findings from the logistic regression analysis showed that the marital status and occupation variables did not have a significant relationship with knowledge and satisfaction in both groups. Only the age variable showed a significant relationship with the uptake of long-acting reversible contraception in the control group. The study participants in the control group were 21% (odds = 0.21) less likely to use long-acting reversible contraception than the study participants in the intervention group. Younger adolescents in the control group were more likely to utilize LARC than older adolescents. These findings inform that if family planning programs put great efforts to pregnant adolescents by ensuring that they get the right information at the right time about postpartum family planning, uptake will be improved following childbirth. In addition, we will plan to grasp through a qualitative study using interviews focusing on the social-cultural context of adolescent pregnant mothers related to the decision aid of long-acting reversal contraception.”